# Folate Related Pathway Gene Analysis Reveals a Novel Metabolic Variant Associated with Alzheimer’s Disease with a Change in Metabolic Profile

**DOI:** 10.3390/metabo12060475

**Published:** 2022-05-24

**Authors:** Jaleel Miyan, Charlotte Buttercase, Emma Beswick, Salma Miyan, Ghazaleh Moshkdanian, Naila Naz

**Affiliations:** 1Division of Neuroscience & Experimental Psychology, Faculty of Biology, Medicine & Health, The University of Manchester, 3.540 Stopford Building, Oxford Road, Manchester M13 9PT, UK; j.miyan@manchester.ac.uk (J.M.); charlotte.buttercase@student.manchester.ac.uk (C.B.); Salma.Miyan@stu.mmu.ac.uk (S.M.); G_moshkdanian@yahoo.com (G.M.); 2LifecodeGX Ltd., 409 Royle Building, London N1 7SH, UK; emma@lifecodegx.com; 3Gametogenesis Research Centre, Kashan University of Medical Sciences, Kashan P.O. Box 8715988141, Iran

**Keywords:** Alzheimer’s disease, cerebrospinal fluid, *MTHFD1*, folate, single nucleotide polymorphism

## Abstract

Metabolic disorders may be important potential causative pathways to Alzheimer’s disease (AD). Cerebrospinal fluid (CSF) decreasing output, raised intracranial pressure, and ventricular enlargement have all been linked to AD. Cerebral folate metabolism may be a key player since this is significantly affected by such changes in CSF, and genetic susceptibilities may exist in this pathway. In the current study, we aimed to identify whether any single nucleotide polymorphism (SNPs) affecting folate and the associated metabolic pathways were significantly associated with AD. We took a functional nutrigenomics approach to look for SNPs in genes for the linked folate, methylation, and biogenic amine neurotransmitter pathways. Changes in metabolism were found with the SNPs identified. An abnormal SNP in methylene tetrahydrofolate dehydrogenase 1 (*MTHFD1*) was significantly predictive of AD and associated with an increase in tissue glutathione. Individuals without these SNPs had normal levels of glutathione but significantly raised MTHFD1. Both changes would serve to decrease potentially neurotoxic levels of homocysteine. Seven additional genes were associated with Alzheimer’s and five with normal ageing. *MTHFD1* presents a strong prediction of susceptibility and disease among the SNPs associated with AD. Associated physiological changes present potential biomarkers for identifying at-risk individuals.

## 1. Introduction

Early-onset, familial Alzheimer’s disease (AD), with a prevalence of around 1%, is known to be associated with high-penetrance mutations in the genes coding for amyloid precursor protein (APP), presenilin 1 (*PSEN1*), and presenilin 2 (*PSEN2*) [1]. *PSEN1* and *2* cause an impairment in γ-secretase activity and lead to an increase in the ratio of the 2 forms of amyloid beta, Aβ1-42: Aβ1-40 leading to AD with an early age of onset spread from 25 to 65 years of age. APP mutations result in early onset AD between 35 and 65 years of age. By contrast, late-onset AD is multifactorial, with many genetic risk factors, including *APOE4*, the highest risk factor for AD, environmental, nutritional, metabolic, and lifestyle factors. No causative genes have been identified for late-onset AD, but non-coding genetic errors have been suggested [2], as well as heritable and non-heritable epigenetic changes, as potential disease onset mechanisms [3], some of which may be offset through nutrition and diet [4,5]. Metabolic disorders have recently been highlighted as a potential cause and target for AD, including insulin signalling dysfunction and brain glucose metabolic disturbances, the latter suggested as hallmarks of AD and underlie the proposition that AD should be regarded as type III diabetes, specifically affecting the brain.

Susceptibility genes for late-onset AD have been identified through genome-wide association studies (GWAS), as well as by deduced candidate genes. These genes are from many different pathways, including lipid metabolism (*APOE*), sortilin-related receptor-1 (*SORL1*), ATP-binding cassette subfamily A member 7 (*ABCA7*), clusterin (*CLU*), immune system and inflammation, including genes coding for complement C3b/C4b receptor 1 (*CR1*), CD33 antigen, membrane-spanning 4-domains, subfamily A member (*MS4A*), triggering receptor expressed on myeloid cells 2 (*TREM2*), member of the major histocompatibility complex class II HLA-DRB5/HLA-DRB1, SH2-containing inositol 5-phosphatase 1 (*INPP5D*), and/or endosome cycling (genes coding for bridging integrator protein-1 (*BIN1*), CD2-associated protein (*CD2AP*), phosphatidylinositol binding clathrin assembly protein (*PICALM*), ephrin type-A receptor 1 (*EPHA1*)) [1]. Using a pathway approach, Novikova et al. [2] found that many genes identified by GWAS were associated with more than a single pathway and identified myeloid cell function (i.e., innate immunity), endocytosis, and phagocytosis as well as lipid metabolism affected by the same genes associated with AD. These functional associations, they argued, are more informative when the multifactorial nature of the disease is appreciated, as well as highlighting potential higher-level primary faults, in this case, myeloid cell function that would impact the microglia of the brain. They went on to show how transcriptome-wide association studies (TWAS) can add to GWAS to identify the potential causality of disease. This is a powerful new approach that may indeed identify genetic risk factors more accurately, as well as potential genetic causes for AD.

Apart from gene associations being proposed as risk factors, including most notably *APOE4*, no gene has been demonstrated to cause late-onset AD in humans or animals. Thus, it seems likely that genetic susceptibilities are triggered by environmental, or other insults individuals may be exposed to throughout their lives, including perhaps most notably, head injury [6,7,8]. New approaches are needed to understand the causes of the aetiology that may lead to dementia or Alzheimer’s to find effective preventatives or treatments. One of these may be functional genomics, as described above, to identify susceptibilities and understand how these are triggered into disease states. Another would be epigenetics, which may uncover later life effects of environmental factors that impact the methylation/acetylation pathways, histone modifications, microRNAs, and other non-coding RNAs species [3,9,10,11]. This would be useful together with GWAS and TWAS, as levels of RNA may indicate levels of protein synthesis but do not account for methylation that is needed by many proteins and lipids for functional states.

For late-onset AD, potential physiological changes should be considered that might precipitate the condition in genetically, or otherwise susceptible individuals. One of the critical physiological processes is the reported observations of changes in cerebrospinal fluid (CSF) output, flow, and drainage. CSF is actively secreted into the ventricles of the brain by the choroid plexus at a rate of 03 mL/min in humans. The choroid plexus is secretory epithelial tissue emanating from the ependymal layer of the brain and projecting into the ventricles. These structures pump CSF into the ventricles, against pressure, and produce a flow of fluid through the ventricles and then into and around the subarachnoid space around the outside of the brain and spinal cord. Fluid drainage is critical as the choroid plexus secretes 4–5 times the fluid volume that can be held in the fluid pathway. Drainage occurs at the facial lymphatics via the arachnoid granulations and villi projecting into the superior sagittal sinus and the perivascular glymphatic pathways. In hydrocephalus, CSF flow and drainage are affected and result in fluid accumulation and rising intracranial pressure, leading to neuropathology and death if not treated. In congenital hydrocephalus, composition changes in CSF deprive cells of folate [12,13], resulting in cell cycle arrest and deficient development [12,14,15]. Different cerebral conditions also have evidence of fluid accumulation associated with severity, including psychosis [16,17], Schizophrenia [18], bipolar [19], autism [20,21,22], and also perhaps hypo-myelination [23]. Although CSF output from the choroid plexus has been reported to decrease with age and dementia [24], CSF drainage, through surgical implantation of a shunt, has produced promising benefits for patients [25], while others have reported raised intracranial pressure [26] as well as enlarged ventricles [25], suggesting a CSF drainage issue. Disease severity associated with increased ventricular enlargement in AD [27,28,29,30] supports a view that a CSF drainage obstruction may be operating in these patients and that shunting, in restoring optimal drainage, improves outcomes. Based on our hypothesis, CSF drainage obstruction, including reduced glymphatic pathway capacity [31,32,33,34,35,36,37,38], may produce cerebral folate issues that would add to the pathophysiology associated with AD.

These physiological effects would be greatly exacerbated by the loss of function in key folate enzymes and transporters, resulting in poor development, function, and plasticity of the cerebral cortex [39,40,41]. For example, some severe neurological conditions are caused by autoantibodies, or abnormal genes for folate receptor alpha (FOLR1) that block the transfer of folate from the blood into the brain, resulting in a severe cerebral folate deficiency. This is associated with poor brain development in autism and increasingly severe neurological signs and symptoms after birth [42,43,44,45,46,47,48,49]. Cerebral folate deficiency syndrome (CFD) is associated with other severe neurological conditions, including spasticity [42,44,45,46,47,50,51], epilepsy [52], extreme behavioural abnormalities [53], encephalopathy [45] and others. CFD is common in children but has also been recognised in adults, indicating that it may develop later in life [54,55,56]. CFD has various other causes, including genetic changes [42,57,58,59,60,61,62,63], potential blockade of folate receptor alpha by high-dose folic acid [64], or autoimmunity [42]. Similar findings have been reported for schizophrenia, including CSF accumulation with enlarged brain ventricle [65,66,67,68] and folate receptor autoantibodies with cerebral folate deficiency [69,70].

Thus, it seems to us that CSF drainage insufficiency, resulting from multiple different causes, would lead to cerebral folate deficiency/imbalance, while at the same time failing to remove toxic molecules, including amyloid. It is also possible that rising amyloid levels could result in CSF drainage compromise through toxic effects on draining cells in the subarachnoid spaces. In either case, the situation would be greatly exacerbated. These possibilities should be investigated as a potential mechanism for various conditions of cerebral cortical malfunction as well as neurodegeneration leading to dementia, including Alzheimer’s disease. 

Nutrigenomics is a relatively new field of research that looks at individual susceptibilities based on analyses of genes involved in metabolic pathways and how they might respond to diet and environmental factors [4,71]. As a first step, in a case-control study design, we aimed to investigate whether any single nucleotide polymorphisms affecting folate and associated metabolic pathways were significantly associated with AD, which might have formed a susceptibility to AD triggered by other (environmental) factors acting on the susceptible metabolism. 

## 2. Results

### 2.1. Folate Metabolism

Figure 1 shows the inter-relationships between the folate metabolic cycle, DNA synthesis, methylation pathway, and neurotransmitter and nitric oxide synthesis. From this, it is clear that errors or issues with folate metabolism can have severe consequences for brain function through the synthesis of neurotransmitters, specifically biogenic amines and acetylcholine, the production and maintenance of cells, cardiovascular and neurovascular health, etc. 5-methyl tetrahydrofolate (5mTHF) is the main dietary form of folate and is the usual entry point into folate metabolism from where it feeds 2 pathways. The dihydro-tetrahydrobiopterin (BH2-BH4) cycle produces tetrahydrobiopterin from 5mTHF that is required for nitric oxide synthesis, linked to cardiovascular health, and biogenic amine synthesis, producing some of the key neurotransmitters of brain functions, including cognition, learning, memory, attention, mood, and sleep. 5mTHF also passes through a rate-limiting step, transferring its methyl group to vitamin B12 that then methylates homocysteine to methionine, producing tetrahydrofolate (THF) that feeds into other parts of the metabolic cycle. Methionine is converted to s-adenosyl methionine (SAM), the universal methyl donor involved in most methylation reactions. 

THF is a central hub of folate metabolism. It can produce 5, 10-methylene THF through the conversion of serine to glycine. This can cycle back to 5mTHF or convert to dihydrofolate (DHF), giving up its methylene to thymidylate synthase that produces pyrimidines, key nucleotides in DNA, and RNA synthesis. DHF is hydrolysed back to THF. THF can also receive a formyl group through the conversion of formate from blood plasma that also exists in CSF [72], or through the metabolism of 5-formyl-THF. This is mediated by methylene-THF-dehydrogenase 1 (MTHFD1), which also converts the product, 10-formyl THF, to 5, 10-methenyl THF. MTHFD1 also acts to balance the 2 halves of folate metabolism through interconversion of 5, 10-methenyl and 5, 10-methylene THF. These 3 reaction steps, mediated by MTHFD1, are known as the long route back to 5mTHF, while MTHFR mediates the final step from 5, 10-methylene THF back to 5mTHF for both the long route and short route. 10-formyl-THF dehydrogenase (FDH) acts as a buffer to maintain a pool of the reactive THF, as it is known to bind tighter to this product than to the substrate 10-formyl-THF [73]. This would also have the effect of depleting 1-carbon availability in the presence of high levels of FDH that would decrease 10-formyl-THF levels and prevent downstream conversions, including purine synthesis. This would explain cell cycle arrest in cancer and other cells produced by elevated levels of FDH [74].

Figure 1 also shows the entry point of folic acid, a synthetic, stable form of folate. At high doses of folic acid, there would be an overload of the system with increased DHF, perhaps slowing down/inhibiting conversion of 5, 10-methylene-THF to DHF and thereby decreasing the production of pyrimidines. Increasing THF levels through this pathway might also have a knock-on effect in slowing down the methylation of homocysteine to methionine. Further details are provided in the figure. High-dose folic acid supplementation has been linked to abnormal folate metabolism and issues related to folate deficiencies [75,76,77] although some caution is required in the interpretation of these studies and conclusions [78]. In our previous research on neonatal hydrocephalus, we also found a negative effect of folic acid in precipitating hydrocephalus in a susceptible rat model of this congenital condition [12]. Folate, but perhaps not folic acid, is thus a vital metabolite, and its metabolism is essential to many associated pathways involved in the development, maintenance, and healthy functionality of tissues, particularly the brain. Up or downregulation of any of the enzymes involved would have significant effects, as would genetic errors affecting the function of these proteins. 

Table 1 gives details of the cases/samples used in the study, while Table 2 analyses clinical and phenotypical characteristics, highlighting the differences between the 2 groups of patients. Interestingly, the AD group is significantly younger than the normal ageing group, with a mean age of 75.2 vs. 86.5. Thal phase, CERAD, score, and Braak stage all accurately associate with the AD group [79] with an additional highly significant association of the *APOE* genotype. Brain weight is significantly lower in AD, with a mean weight of 1133.5 g vs. 1291.9 g. Although the mean postmortem delay is different between groups, this, along with brain pH, does not show significance.

### 2.2. SNPs in Neural-Related Genes

Among the 44 SNPs studied, the genotypes at two SNPs, COMT (rs4633) and COMT (rs4680), emerged as strong predictors for AD. The CC genotype at COMT (rs4633) (OR: 14.99, 95% CI: 2.024–111.2), and GG at COMT (rs4680) (OR: 12.00, 95% CI: 1.581–91.084) showed significant differences between the control and AD patients. 

Allelic distribution revealed that at COMT (rs4633), the C allele was highly prevalent in AD patients, while the T allele was more common among the normal group (*p* = 0.008; OR: 3.24; 95% CI: 1.415–7.398). At COMT (rs4680), allele A was more prevalent among the normal subjects and G among the AD patients (*p* = 0.024; OR: 2.78; 95% CI: 1.217–6.367). Interestingly, at CYP2D6 (rs16947), the G allele was more prevalent among the AD patients and an allele among the normal subjects (*p* = 0.02; OR: 2.91; 95% CI: 1.231–6.895) (Appendix A).

In the multivariable model, three SNPs emerged as strong predictors of AD (COMT –rs4680 omitted due to collinearity) (Table 1). The contribution of ADRB1 (rs1801253) was observed to be highest in disease status (OR: 9.61). The risk of SNPs in the multivariable analyses was best explained by the dominant model (Table 3).

### 2.3. SNPs in Folate and Methylation-Related Genes

Among the 37 SNPs studied, the genotypes at four SNPs, i.e., *MTHFD1* (rs1076991), *MTHFD1* (rs2236225), *SOD2* (rs2758331), and *APOE* genotype, were found to be the strong predictors of AD. Accordingly, the TT genotype at *MTHFD1* (rs1076991) (OR: 14.00, 95% CI: 2.079–94.24), GG at *MTHFD1* (rs2236225) (OR: 2.06, 95% CI: 0.455–9.304), CC at *SOD2* (rs2758331) (OR: 10.00, 95% CI: 1.594–62.73), and E3/E4 at *APOE* (OR: 16.50, 95% CI: 2.818–96.61) showed statistically significant differences among the controls and AD patients (Appendix A). Differences in the allelic distribution at 5 loci, i.e., *MTHFD1* (rs1076991), *SUOX* (rs705703), *SOD2* (rs2758331), *SOD2* (rs4880), and *APOE*, were statistically different (Appendix A). In the multivariable model, five SNPs appeared to be the strongest predictors of AD (Table 3). 

The highest odds were observed for *CBS* (rs234706) and *MTHFD1* (rs1076991) (OR: 95.12 and 67.38, respectively). The risk of SNPs was most significant under the additive model (Table 4). 25 gene variants were identified that were significant in any of the three tests with all significant at *p* ≤ 0.05 and many at the much higher significance of *p* ≤ 0.01 or higher (bold *p*-values in Table 5). Only 12 were significant using Mann–Whitney U tests (Appendix A), while more were significant in either of the Chi Squared tests. Some were significant across all tests (Table 5). The most significant finding was that ApoE4 was associated with AD. Even though ApoE4 is known in the literature to have a 40% risk of the disease, the current finding is surprising in showing a 70% association in a small number of individuals picked for disease severity. 

Importantly for our hypothesis, 2 folate-related genes were found to be significantly associated with AD, methylene tetrahydrofolate dehydrogenase 1 (*MTHFD1*) and methylene tetrahydrofolate reductase (*MTHFR*), with *MTHFD1* significant on all tests and *MTHFR* significant only on a Chi, Squared test in which positive and neutral variants were grouped and tested against negative variants. Table 5 also shows the direction of the association, i.e., whether associated with AD or with normal ageing. Gene SNPs associated with normal ageing may provide some protection from AD.

Other SNPs associated with AD are involved in monoamine transport at synapses (*SLC18A10, SLC6A2*) as well as involved in detoxification of xenobiotics and sulphites (*CYP2D6, SUOX*). Those associated with normal ageing and not AD, which may therefore be protective against AD, are involved in monoamine metabolism, methylation, and signalling (*MAOA, ADRB1, COMT*), and detox pathways (*CYP2D6, SOD2, GSTM1*). There are also SNPs involved in thyroid hormone activation and transport (*DIO2, SLC01C1*) and neurotransmitter receptors (*5-HT2A, DRD2*). The remaining SNPs, only significant on Chi Squared tests where negative and neutral SNPs are grouped, are associated with AD and are involved in methylation, including betaine homocysteine methyl transferase (*BHMT*), cysteine beta-synthase (*CBS*), and glutathione S-transferase P1 (*GSTP1*).

### 2.4. Changes in the Metabolic Profile Associated with Folate Gene SNP

We measured folate metabolites and enzymes in tissue lysates of normal and Alzheimer’s disease individuals with negative SNPs in *MTHFD1* and/or *MTHFR* and compared these to normal and Alzheimer’s disease individuals with normal or neutral SNPs in these genes. The individuals, their genotypes, and results of the analysis are shown in Table 6 and Table 7, with the data shown in graphical forms in Figure 2 and Figure 3.

There was no significant difference in tissue folate levels (Figure 2) although the controls had a non-significant reduced level compared to the other groups. We, therefore, used the average folate level to calculate the fold levels of the other metabolites and enzymes (Figure 3). In the severe Alzheimer’s cases that have negative SNPs in *MTHFD1*, there is a significant increase in glutathione that is seen in the severe cases without these SNPs. There was no effect of the negative SNPs on the levels of either *MTHFD1* or *MTHFR*. However, in severe cases with normal or neutral SNPs, there is no rise in glutathione, but there is a significant rise in MTHFD1.

There is also a significant rise in MTR in severe cases, both with and without negative SNPs, relative to controls and a small but non-significant (*p* ≤ 0.06) increased MTR in severe cases without negative SNPs compared to severe cases with negative SNPs.

## 3. Discussion

There is an assumption in the genetics and nutrigenomics literature that adverse gene SNPs have a negative effect on protein function and thus a knock-on effect on the processes they are involved in. In this study, we focused on the genes involved in folate metabolism, methylation, and neurotransmitter synthesis, and also included *APOE* genotyping. Surprisingly, we found a 70% association of *APOE4*, rather than the predicted 40%, with AD further reinforcing its high-risk factor status and also indicating potential direct involvement in the condition in the severe cases used in this study. Several recent studies highlight the role of ApoE in several critical processes, including involvement in amyloid plaque and neurofibrillary tangle formation, insulin resistance, decreased amyloid clearance, mitochondrial dysfunction, and autophagy [5,80]. Suggestions have been made for targeted drugs [80] and nutritional and lifestyle changes [5] aimed at ApoE4 processes and pathways to prevent or treat the disease. The second most significant association with AD was *MTHFD1* SNP rs1076991. This was highly significant in all tests, and thus we can suggest must be a significant risk factor for late-onset AD. Others have found a weak association between a different *MTHFD1* variant, SNP rs2236225, and early-onset AD [81,82]. We found a highly significant association of this variant with AD only in a Chi-Squared test, where negative and neutral variants were grouped together and tested against positive variants. There was no significance in the Mann–Whitney U test or the alternative Chi-Squared test. Therefore, we can agree with the studies demonstrating the weak association of this variant but have found a very significant association with the other variant of *MTHFD1*, which is a novel finding of this study. MTHFD1 is a critical folate enzyme involved in 3 parts of folate metabolism (Figure 1) and 4 enzymatic profiles: (i) methylenetetrahydrofolate dehydrogenase (NADP+ dependent) 1, (ii) methenyltetrahydrofolate cyclohydrolase, (iii) formyltetrahydrofolate synthetase, and (iv) C-1-tetrahydrofolate synthase, reflecting its importance to the folate metabolic cycle. Thus, a negative variant of *MTHFD1* should have a remarkable effect on folate metabolic balance, decreasing the 5mTHF pool and potentially leading to homocysteine and formate toxicity and errors/reduced DNA synthesis and repair. Similarly, a negative variant of *MTHFR* should result in raised homocysteine levels that would exacerbate neurodegeneration and increase the risk of developing AD [83]. In the cases studied here, *MTHFD1* is very significantly associated with AD, while *MTHFR* is probably only weakly associated, as already reported for this variant [84] by contrast to other studies finding 1–3 additional variants of this gene associated with AD [83,85,86]. We found a significant increase in glutathione in severe Alzheimer’s cases with negative variants in *MTHFD1* and/or *MTHFR* compared to those with positive or neutral variants (Figure 2). We surmise that raised homocysteine, resulting from failure to regenerate 5-methylTHF, is being shunted to SAM and glutathione to prevent the toxic build-up of homocysteine. Interestingly, this was not seen in severe cases with positive or neutral variants in *MTHFD1* and/or *MTHFR*. In these cases, we found a significant increase in MTHFD1 and MTR perhaps in response to raised homocysteine and to increase methylation. This would also fuel the hypermethylation seen in the Alzheimer’s cortex and reported by us and others (Naz et al., 2022, in revision) and may explain the reduced SAM we report here. Additionally, we found no significant difference in tissue levels of folate, indicating that the changes seen are likely to be a response to ineffective gene products and/or to physiological changes in metabolism rather than folate supply. These significant changes in metabolic profile may provide early markers of at-risk individuals who may therefore be protected by appropriate metabolic support, including perhaps folate supplements [12,87,88,89].

The other gene variants associated with AD are involved in monoamine neurotransmitter delivery to, and reuptake into synapses, and in detoxification from xenobiotics and sulfites. 2 genes involved in the methylation pathway are weakly associated with AD, finding significance in only one of the Chi Squared tests. These are involved in the conversion of homocysteine to methionine (*BHMT*) and in the generation of cysteine from homocysteine in the pathway to glutathione (*CBS*). The remaining significant associations are with normal ageing, indicating a possible protective effect of these gene variants. These are involved in receptors for and breakdown of monoamines, as well as in detoxification through breakdown of various drugs and environmental toxins. Interestingly, *COMT* variant rs4633 has been associated with Alzheimer’s [90] while in the current study it is clearly associated with normal ageing. Other variants of *COMT* have been found to be not associated with Alzheimer’s or other psychiatric conditions [91,92,93] indicating that *COMT* may be associated with other related factors rather than directly to disease aetiology or progression.

We propose that one trigger for the onset and severity of Alzheimer’s may be a physiological change associated with a cerebral CSF drainage issue and a consequential cerebral folate issue. This is reflected in the fact that the severity of this, and other cerebral conditions, is associated with increasing fluid accumulation and ventricular dilation. Life events that decrease drainage capacity might include infection, inflammation, and trauma or accelerated cell loss in these susceptible ageing individuals as well as due to toxic levels of amyloid, e.g., as found in sleep deprived individuals. Strategies to maintain drainage, and perhaps increase drainage may therefore present an effective target for prevention and treatment of this condition. Changes in metabolic profiles, as found in this study, may also provide early markers of at-risk individuals.

## 4. Material and Methods

Brain tissue: All brain tissues were supplied from the Manchester Brain Bank under their ethical approval (09/H0906/52+5 and 19/NE/0242) for the collection and use of human tissue in ageing research. In this study, we were provided with frozen unfixed tissue from the occipital cortex that included the full thickness of the cortex from the pia to the ventricular ependymal. CSF samples were also provided from the same brains, which were collected postmortem. Information on the cases used is provided in Table 1. Only individuals who were clearly normal, based on both pathology and clinical observations prior to death, or who were clearly suffering severe Alzheimer’s disease were included in this study. 25 individuals from each category were analysed. 

The cerebral cortical plate was dissected and then sent to a commercial company, LGC Genomics, for quality checking, further processing, and SNP analysis. They extracted DNA from the tissue using their in-house LGC Kleargene extraction chemistry. Genotyping was performed by LGC genomics using their in-house competitive allele specific PCR (KASP) technique. Genotype data were sent to LifecodeGX Ltd. to input them into their bespoke software that matches SNPs to specific metabolic pathways. The software also categorised SNPs according to their functional effect based on literature reviews, namely beneficial, neutral, or harmful (see https://www.lifecodegx.com/, accessed on 19 May 2022, for details and references). These data were then tabulated in Microsoft Excel. Heat maps were generated in Excel to give a pictorial representation of the SNP data (Supplementary data). Statistical analyses were carried out using GraphPad Prism and STATA (11.0). Genotypic and allelic frequencies of SNPs were calculated and compared between normal subjects and AD patients. Categorical variables were given as absolute and relative frequencies (no., %age). The significance of the differences from random distribution was estimated through Chi-square, Yates’ corrected Chi-square, and Fisher’s exact tests. The quantitative variables were expressed as mean ± SD (standard deviation), and T-test statistics were employed for comparison. At each SNP, odds ratios (OR) and respective 95% confidence intervals (95% CI) and univariable regression were calculated while keeping the most common homozygous genotype as a reference. Multivariable regression was performed to identify the combination of SNPs associated with AD. For this, a stepwise logistic regression model was applied by including all SNPs one by one and eliminating the non-significant ones. To predict the most plausible inheritance model, significant SNPs in multivariable regression were also assessed under dominant, recessive, and additive models. The level of significance for the *p*-value was <0.05 for all tests.

## Figures and Tables

**Figure 1 metabolites-12-00475-f001:**
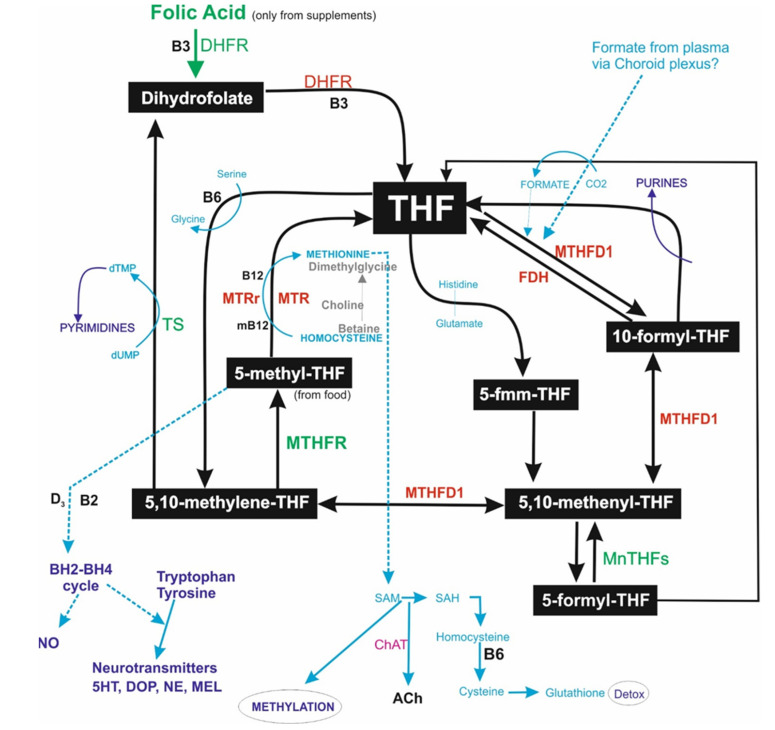
Diagram of the folate metabolic cycle and links to other pathways 5-methyl-THF is the major species of folate derived from food and forms the recycling point for folate metabolism. It forms the rate-limiting step, through the action of methionine synthase (MTR), in the methylation of vitamin B12 and thus the rate-limiting step for the methylation of homocysteine to methionine. Thus, it is critical to folate metabolism generally and to the production of S-adenosyl methionine (SAM), the universal methyl donor for methylation reactions. MTHFD1 is a multi-role enzyme involved in three reactions in the folate pathway, forming a long route back to 5-methyl-THF. Tetrahydrofolate (THF) can be recycled back through 5, 10-methenyl THF, forming a short route and requiring B6 and serine to glycine reactions. Both long and short routes require methyleneTHF reductase (MTHFR) for the final step to 5-methyl-THF. MTHFD1 is further involved in recycling 10-formyl-THF to THF, fuelling purine synthesis. 5, 10-methenyl THF can also be reduced to dihydrofolate, fuelling pyrimidine synthesis. Other pathways include 5-methyl-THF feeding directly into biogenic amine and nitric oxide synthesis through the BH4 cycle, methionine feeding directly into the methylation pathway, and acetylcholine synthesis. Folic acid is an artificial substance that enters the folate cycle without any 1 carbon moiety to supply to the metabolic process and so acts to dilute the 1 carbon pool, as well as having other negative effects at higher doses (see text).

**Figure 2 metabolites-12-00475-f002:**
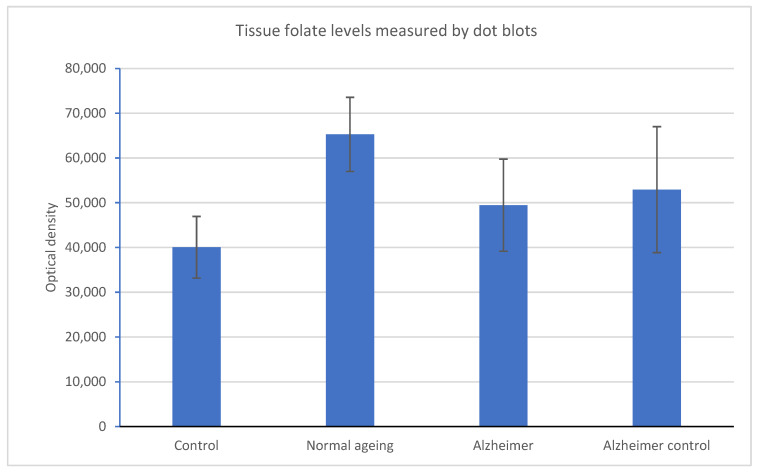
Comparison of negative and positive gene SNPs on tissue folate levels in the groups used in the study metabolic profiles. Folate levels were not significantly different between controls and other groups with and without mutant SNPs in MTHFD1 and/or MTHFR. The controls have a lower level of folate than the other groups, although this is not significant.

**Figure 3 metabolites-12-00475-f003:**
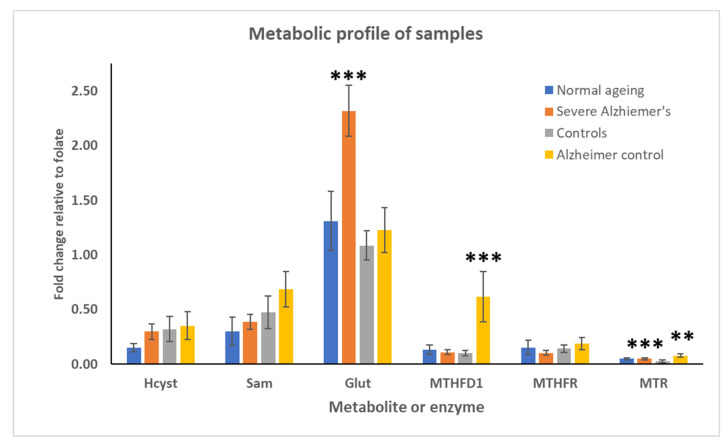
Metabolic profiles of samples analysed for key metabolites and enzymes plotted as fold of folate levels since this was the only consistent measure between the cases. Glutathione is significantly raised in Severe Alzheimer’s with mutant gene SNPs for MTHFD1 and MTHFR. This is not seen in any other group, including Alzheimer’s without mutant SNPs. In these latter cases, we see significantly raised MTHD1, which is not a mutant. In both cases, we see potential protective mechanisms against raised homocysteine levels, which are elevated in all cases except normal ageing, although none are significant. MTR is also elevated in both affected and control Alzheimer’s cases. This is not mutated, so it is also involved in metabolising 5-methyl-THF to methylate homocysteine to methionine and thus feeding the methylation pathway. Together, the increased glutathione and MTR would act to keep homocysteine levels low. Significance is indicated by stars: ** significant at 0.01, *** significant at 0.001.

**Table 1 metabolites-12-00475-t001:** Cases used in the study.

Normal Ageing Cases									
Case No.	MRC ID	Gender	Age at Death	Braak Stage	PMD (h)	Clinical Diagnosis	Pathological Diagnosis 1	Pathological Diagnosis 2	APOE
DPM12/11	BBN_3478	M	54	0	37	control	normal brain		33
DPM14/04	BBN_19634	F	87	0-I	24	Normal	Age changes only		34
DPM14/08	BBN_20005	M	85	0-I	98	Normal	Age changes only	moderate SVD	33
DPM14/20	BBN_21003	F	90	0-I	39	normal	Age changes only		33
DPM14/46	BBN_24316	F	94	0-I	111	control	age changes only	mild SVD	23
DPM16/29	BBN005.29063	M	69	0-I	53	Control	Normal for age		24
DPM18/03	BBN005.32560	M	88	0-I	39	Control	Normal for age		33
DPM14/09	BBN_20006	M	84	I	69.5	Normal	Age changes only	moderate SVD	33
DPM17/09	BBN005.30100	F	88	I	52.5	Control	Normal for age	ARTAG, possible PART	23
DPM17/34	BBN005.31485	M	89	I	125	Control	Normal for age	Incidental Lewy bodies?	23
DPM09/31	BBN_3396	F	94	I-II		cognitively normal/stroke	Age changes only	mild to moderate SVD	33
DPM12/09	BBN_3476	F	87	I-II	87	cognitively normal	mild AD pathology in temporal lobe		33
DPM13/35	BBN_15591	F	76	I-II	47	normal	mild AD changes in temporal lobe	very mild CAA, moderate SVD in BG	33
DPM14/11	BBN_20195	M	91	I-II	43.5	Normal	mild SVD		33
DPM15/01	BBN_24368	M	90	I-II	156	control	Age changes only		33
DPM16/11	BBN005.28403	M	77	I-II	63	Control	Mild temporal tau, possible PART		33
DPM16/31	BBN005.29168	M	90	I-II	155	Control	Normal for age	Mild SVD	33
DPM17/15	BBN005.30170	M	90	I-II	125	Control	Normal for age	Incidental Lewy bodies?	33
DPM17/36	BBN005.32382	F	94	I-II	70	Control	Age changes only		33
DPM18/11	BBN005.32822	F	90	I-II	143	Control	Age changes only	Possible ARTAG	33
DPM11/06	BBN_3446	F	92	II	37	cognitively normal	Age changes only	mild SVD	34
DPM11/25	BBN_3463	M	89	II	27	Control	Age changes only		33
DPM11/29	BBN_3467	M	89	II	123	cognitively normal	Age changes only	mild SVD	33
DPM15/28	BBN_25917	F	91	II	133	Control	Age changes only	Cerebral infarction	23
**Severe Alzheimer’s disease cases**									
Case No.	MRC ID	Gender	Age at death	Braak stage	PMD (h)	Clinical diagnosis	Pathological diagnosis 1	Pathological diagnosis 2	APOE
DPM16/16	BBN005.28547	F	81	V	176	AD	AD (Braak V)	V.severe CAA	34
DPM12/01	BBN_3469	M	67	V-VI	84	Dementia	Alzheimer’s disease	mild SVD	34
DPM12/32	BBN_9480	M	73	V-VI	36	Alzheimer’s disease	Alzheimer’s disease		33
DPM13/09	BBN_11027	F	85	V-VI	73	Alzheimer’s disease	Alzheimer’s disease	moderate to severe SVD, v. Mild DLB	34
DPM13/10	BBN_11028	F	85	V-VI	24	dementia	Alzheimer’s disease	Mild CAA	34
DPM13/45	BBN_19208	M	78	V-VI	138	Alzheimer’s disease	Alzheimer’s disease		33
DPM14/10	BBN_20007	F	78	V-VI	70	Alzheimer’s disease	Alzheimer’s disease	CAA with capillary involvement	44
DPM14/50	BBN_24361	F	63	V-VI	54	Alzheimer’s Disease	Alzheimer’s disease	moderate SVD	44
DPM15/02	BBN_24373	M	78	V-VI	173	Alzheimer’s Disease	Alzheimer’s disease	sec TDP-43 proteinopathy, incidental Lewy bodies?	44
DPM17/37	BBN005.32384	F	90	V-VI	76	AD	Alzheimer’s disease	Possible AGD	34
DPM11/28	BBN_3466	F	71	VI	64	Alzheimer’s disease	severe Alzheimer’s disease		44
DPM12/03	BBN_3470	M	72	VI	81	Alzheimer’s Disease	Alzheimer’s disease		34
DPM12/05	BBN_3472	M	73	VI	107	Alzheimer’s Disease	Alzheimer’s disease	mod SVD	44
DPM14/30	BBN_23794	F	70	VI	89	dementia, learning difficulty	Alzheimer’s disease		44
DPM14/31	BBN_23803	M	64	VI	98.5	Alzheimer’s Disease	Alzheimer’s disease	moderate SVD	34
DPM15/48	BBN005.26301	F	81	VI	98	Dementia	AD (Braak VI)	Secondary TDP-43	34
DPM16/10	BBN005.28400	F	59	VI	87	AD	Alzheimer’s disease		24
DPM16/40	BBN005.29461	M	82	VI	25.5	AD	Alzheimer’s disease	Moderate CAA	34
DPM18/12	BBN005.32823	M	70	VI	120.5	AD?	Alzheimer’s disease	Moderate SVD	33
DPM18/39	BBN005.35131	F	75	VI	127.5	Dementia	Alzheimer’s disease		34
Case No.	MRC ID	Gender	Age at death	Braak stage	PMD (h)	Clinical diagnosis	Pathological diagnosis 1	Pathological diagnosis 2	APOE
DPM16/16	BBN005.28547	F	81	V	176	AD	AD (Braak V)	V.severe CAA	34
DPM12/01	BBN_3469	M	67	V-VI	84	Dementia	Alzheimer’s disease	mild SVD	34
DPM12/32	BBN_9480	M	73	V-VI	36	Alzheimer’s disease	Alzheimer’s disease		33

**Table 2 metabolites-12-00475-t002:** Clinical and phenotypic differences between Alzheimer’s disease (AD) patients and normal ageing control samples.

Variable	Control	AD Patients	Total	*p*
Gender				
Male	14	12	26	
Female	11	13	24	0.571
Sum	25	25	50	
Age at death (mean + SD)	86.5 (9.1)	75.2 (9.0)	80.9 (10.6)	0.0001
Pathological diagnosis 1				
Age changes only	22	0	22	
Mild temporal lobe pathology	3	0	3	
AD typical	0	17	17	
AD and dementia	0	6	6	
AD and learning difficulty	0	1	1	
AD atypical	0	1	1	
Pathological diagnosis 2				
Mild SVD	4	1	5	
Moderate SVD	4	4	8	
Severe SVD	1	1	2	
Cerebral infarction	1	0	1	
Incidental Lewy bodies	2	1	3	
Possible ARTAG	2	1	3	
CAA	0	4	4	
Secondary TDP-43	0	5	5	
Thal phase (*n* = 29)				
0–1	9	0	9	
2–3	5	3	8	
4–5	0	12	12	0.0003
CERAD score (*n* = 31)				
0	6	0	6	
A	7	0	7	
C	0	18	18	<0.0001
Braak stage				
0–I	10	0	10	
I–II	15	0	15	
V–VI	0	25	25	<0.0001
*APOE genotype*				
2/3	4	0	4	
2/4	1	1	2	
3/3	18	6	24	
3/4	2	11	13	
4/4	0	7	7	0.0001
Brain weight, gm (mean ± SD)	1291.9 (184)	1133.5 (130)	1211.0 (176)	0.0014
PMD (mean ± SD)	88.8 (45.6)	100.9 (56.4)	92.5 (51.6)	0.2478
Brain pH (mean ± SD)	5.9 (0.3)	6.2 (0.5)	6.1 (0.4)	0.0617

**Table 3 metabolites-12-00475-t003:** Univariate and multivariate logistic regression of predictors of AD status.

SNPs	Univariate			Multivariate		
	OR	95% CI	*p*-Value	OR	95% CI	*p*-Value
Nervous system related						
*COMT (rs4633)*	3.80	1.418–10.19	0.008	5.50	1.412–21.39	0.014
*COMT (rs4680)*	3.41	1.256–9.251	0.016	1.00	(omitted)	
*CYP2D6 (rs16947)*	2.44	1.077–5.544	0.033	3.64	1.135–11.69	0.030
*ADRB1 (rs1801253)*	2.82	0.976–8.133	0.055	9.61	1.781–51.87	0.009
_cons				0.00	0.000–0.027	0.002
Methylation related						
*MTHFD1 (rs1076991)*	3.98	1.550–10.23	0.004	67.38	3.231–1404	0.007
*MTHFD1 (rs2236225)*	1.52	0.726–3.164	0.268	10.50	1.595–69.10	0.014
*MAT1A (rs1985908)*	0.62	0.263–1.448	0.267	0.04	0.004–0.540	0.015
*CBS (rs234706)*	1.90	0.798–4.533	0.147	95.12	3.455–2618	0.007
*APOE*	2.07	1.196–3.592	0.009	4.46	1.193–16.66	0.026
_cons				0.00	5.3 × 10^−8^–0.004	0.006

SNPs that remained significant in multivariate when adjusted for gender and age (except *CYP2D6*-rs16947 and *APOE* which appeared not significant with age).

**Table 4 metabolites-12-00475-t004:** Significant predictors of AD status assessed in various inheritance models.

SNPs	Dominant		Recessive		Additive	
	OR	*p*-Value	OR	*p*-Value	OR	*p*-Value
Nervous system related						
*COMT (rs4633)*	22.05	0.017	3.41	0.123	5.50	0.014
*COMT (rs4680)*	1.00	(omitted)	1.00	(omitted)	1.00	(omitted)
*CYP2D6 (rs16947)*	14.92	0.044	3.22	0.076	3.64	0.030
*ADRB1 (rs1801253)*	15.51	0.013	1.00	(omitted)	9.61	0.009
_cons	0.003	0.009	0.39	0.050	0.00	0.003
Methylation related						
*MTHFD1 (rs1076991)*	6.71	0.125	8.22	0.010	483.8	0.026
*MTHFD1 (rs2236225)*	2.19	0.416	3.28	0.149	22.22	0.036
*MAT1A (rs1985908)*	0.50	0.409	1.3 × 10^−7^	0.991	0.073	0.034
*CBS (rs234706)*	4.92	0.071	2.3 × 10^6^	0.992	465.8	0.025
*APOE*	20.45	<0.0001	1.00	(omitted)	161.3	0.022
_cons	0.018	0.033	0.29	0.017	1.9 × 10^−7^	0.024

**Table 5 metabolites-12-00475-t005:** Gene variants are significantly associated with normal ageing and Alzheimer’s disease.

	Mann Whitney, *p* < 0.05 (*p* ≤ 0.01 in bold)		Chi squared (R+Y), *p* ≤ 0.05		Chi squared (Y+G), p ≤ 0.05		EFFECT	
Protein	Genes (variant)	*p* value	Genes	*p* value	Genes	*p* value	AD	N
Apolipoprotein E4 fat metabolism—principle cholesterol carrier in brain supplying neurones via lipoprotein receptors	APOe4	**1.61 × 10^−6^**	APOe4	0.029096332	APOe4	**4.12 × 10^−32^**		
MethyleneTHF dehydrogenase long pathway replenishment of 5mTHF	MTHFD1 (rs1076991)	**0.000982**	MTHFD1 (rs1076991)	**0.010097315**	MTHFD1 (rs1076991)	**4.88 × 10^−8^**		
			MTHFD1 (rs2236225)	**0.000311491**				
MethyleneTHF reductase final step in long and short pathway back to 5mTHF					MTHFR (rs1801131)	0.026992		
synaptic vescile associated monoamine transporter	SLC18A1 (rs1390938)	0.029941			SLC18A1 (rs1390938)	0.004797		
monoamine transporter responsible for reuptake from synapse	SLC6A2 (rs5569)	0.045301	SLC6A2 (rs5569)	0.01562887	SLC6A2 (rs5569)	0.015629		
Cytochrome oxidase involved in metabolism of xenobiotics	CYP2D6 (rs1135840)	0.036104	CYP2C19 (rs4244285)	0.029096332				
Mitochondrial enzyme—sulfite oxidase—detox	SUOX (rs705703)	**0.00987**	SUOX (rs705703)	**0.012419331**				
β-Adrenergic receptor	ADRB1 (rs1801253)	0.032407			ADRB1 (rs1801253)	**0.010097**		
Catechol-O-methyl transferase—degrades monoamines	COMT (rs4633)	**0.002408**	COMT (rs4633)	**2.15** **×10^−5^**	COMT (rs4633)	**0.001832**		
	COMT (rs4680)	**0.005641**	COMT (rs4680)	**0.001155233**	COMT (rs4680)	**0.008119**		
cytochrome P450 Breakdown of medicines	CYP2D6 (rs16947)	**0.0107**	CYP2D6 (rs16947)	**9.00** **×10^−5^**	CYP2D6 (rs1135840)	**0.001063**		
Iodothyronine deiodinase activates thyroid hormone			DIO2 (rs225014)	0.045327562	DIO2 (rs225014)	0.026992		
superoxide dismutase—detox from oxidative products	SOD2 (rs2758331)	**0.004987**	SOD2 (rs2758331)	**0.000221847**	SOD2 (rs2758331)	**0.004267**		
	SOD2 (rs4880)	**0.009887**	SOD2 (rs4880)	**0.000221847**	SOD2 (rs4880)	**0.014306**		
Glutathione S-transferase—detox from drugs, environmental toxins, oxidative stress			GSTM1 (insert/delete)	0.035014981	GSTM1 (insert/delete)	0.035015		
Monoamine oxidase					MAOA (rs6323)	0.019208		
5HT receptor 2A			5-HT2A (rs6311)	**0.002088939**				
Dopamine receptor D2			DRD2 (rs6277)	0.045500264				
			IFN-g (rs2430561)	0.025935446				
iodothyronine deiodinase deiodination of T4			DIO1 (rs2235544)	**0.012419331**				
Solute carrier—high affinity transport of organic anions (e.g. T4 and other hormones) may act at BBB			SLCO1C1 (rs10770704)	0.002199647				
Betaine--homocysteine S-methyltransferase 1 required for Hcyst to Methionine			BHMT (rs567754)	0.043951044				
Cystathionine beta-synthase downregulates methionine by converting HCYst to cycsteine			CBS (rs234706)	0.045327562				
Glutathione S-transferase P-conjugates glutathione to wide range of electrophiles/toxins			GSTP1 (rs1695)	0.041226833				

Gene SNP variants are significantly associated with normal ageing and Alzheimer’s disease. All the gene variants shown are significantly associated with AD, indicated by the red bar, or with normal ageing, indicated by the blue bar, at *p* < 0.05 level. *p*-values less than 0.01 are indicated by the darker grey cells. The Mann–Whitney U test was used to compare the numbers of Red: Yellow: Green cells between normal ageing and severe Alzheimer’s disease. The full details of the statistical tests for all genes are shown in Appendix A. 2 separate Chi-squared tests are also presented to compare differences when yellow is merged with red, or with green. The most significant association is with *APOE4* with *MTHFD1*. These are significant on any test, while *MTHFR* is only significant in Chi-squared, where yellow is merged with green (Y+G). The data are shown in Appendix A. Other details are discussed in the text.

**Table 6 metabolites-12-00475-t006:** Samples used in folate analyses correlated to gene SNP data.

	Normal Ageing				Alzheimer’s					Normal Controls				Alzheimer’s Controls			
SampleIDs Target molecules	12_11	09_31	14_09	14_11	12_01	13_45	14_50	19_29	19_31	11_25	14_08	17_36	17_34	19_04 A	12_05 B	12_32 C	11_28 D
MTHFD1 (rs1076991)	CT	TT	TT	TT	TT	TT	TT	TT	TT	CC	CC	CT	CC	CT	CT	CT	CT
MTHFD1 (rs2236225)	AA	AG	AA	AA	AA	AA	GG	GG	AA	GG	GG	AG	AG	AG	GG	GG	
MTHFR (rs1801131)	TT	GT	GT	TT	GG	GG	TT	GG	TT	GT	GT	TT	TT	TT	GT	GT	GT
MTHFR (rs1801133)	AA	GG	AG	AG	GG	GG	AA	GG	GG	AG	AG	GG	GG	AG	AG	GG	GG
**DOT BLOTS**																	
Homocysteine	9190	10,600	5850	10,900	14,800	12,900	6640	7230	21,100	8490	26,100	3510	9720	14,700	49,000	18,500	157
SAM	7010	27,700	8080	23,100	9330	16,800	15,500	18,200	27,000	6000	38,000	8040	28,000	11,300	68,900	32,100	27,500
Glutathione	58,600	90,200	91,400	78,100	97,600	69,100	160,000	119,000	97,400	25,000	60,600	39,000	43,200	43,800	95,200	36,900	63,800
Folates	75,800	44,100	80,800	60,400	34,800	32,800	88,700	40,400	50,600	19,600	44,700	46,200	49,700	46,000	99,200	31,400	35,100
**WESTERN BLOTS**																	
MTHFD1	8560	9910	2070	9990	5390	5490	4530	3360	4680	873	3540	6200	7650	19,700	9210	36000	28,500
MTHFR	NU	NU	4850	15,000	2690	3130	5110	6980	6030	4850	4420	5290	5630	14,100	3470	7200	6650
MTR	NU	NU	2900	3950	2020	2840	2380	1570	2120	1230	863	717	452	5290	4260	2740	2290

Samples used in folate analyses correlated to gene SNP data. Normal ageing and Alzheimer’s cases with mutant SNPs for *MTHFD1* and *MTHFR* were selected along with individuals from the same groups having no mutant SNPs or only one copy of the mutant SNP (normal and Alzheimer’s controls). Beneath the heat maps are the dot or Western blot intensity data for each of the enzymes or metabolites measured. Red are homozygous for abnormal SNP, yellow are heterozygotes, green are homozygous for normal SNP.

**Table 7 metabolites-12-00475-t007:** *T*-tests of control.

	*p* Values		
	CvN	CvA	CvAC
Hcyst	0.554	0.915	0.303
SAM	0.267	0.737	0.139
Glut	**0.005**	**0.008**	0.098
MTHFD1	0.332	0.931	**0.033**
MTHFR	0.530	0.795	0.297
MTR	0.145	**0.002**	**0.013**
Folate	0.140	0.497	0.502

Tests of significance. Bold values were calculated for each measure against that for tissue folate, which was not significantly different between the groups (see Figure 2) and thus was used as a measure of all other elements. Each metabolite or enzyme was tested against a normal control (C). Glutathione is significantly different in both normal and Alzheimer’s with mutant SNPs but not in Alzheimer’s cases that do not. *MTR* is significantly different in both Alzheimer’s with or without negative SNPs. *MTHFD1* is significantly different only in Alzheimer’s without negative SNPs. These data are shown in graphical form in Figure 2 and Figure 3.

## Data Availability

The data is presented in the paper in tables and Appendix A.

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
