# Peer review of "Folate Related Pathway Gene Analysis Reveals a Novel Metabolic Variant Associated with Alzheimer’s Disease with a Change in Metabolic Profile"

_metabolites, 2022, doi:10.3390/metabo12060475_

Round 1
Reviewer 1 Report
This paper investigates SNPs and some metabolite measurements from the folate pathway with regard to their role in Alzheimer's disease. This is a very interesting and important topic as there is significant evidence of the importance of the role of folate metabolism in neurodegenerative and neurodevelopmental conditions. The paper generally reads well and seems technically sound. Most importantly the findings are significant. The sample size is not large but then this is a limitation of the subject matter (due to the use of cerebrospinal fluid) and cannot easily be addressed.
Some specific comments are given below:
- There are a number of formatting problems that should have been caught while proofreading the submitted paper. For example, Table 1a and b overlap which makes some of the content unreadable; Table 6b uses a much larger font than found anywhere else in the manuscript.
- The paper measures SAM as one of the variables. Do you have the ability to also measure SAH as the SAM/SAH ratio has been found to be different in some other conditions as compared to controls? Given the link between SAM/SAH and DNA methylation this would be an important observation.
- The paper would benefit from having a bit more detail on the analysis used. While the methods are reasonably standard, it would not hurt to provide more information than is currently in the paper.
- The paper contains a number of typographical/grammatical mistakes that should be corrected, e.g., but not limited to, "Alzhiemer" in Figure 2b, "no significantly" instead of "not significantly" in the legend of Table 6, "This was not found to significantly difference" in legend of Table 1, etc.
Author Response
- Q. There are a number of formatting problems that should have been caught while proofreading the submitted paper. For example, Table 1a and b overlap which makes some of the content unreadable; Table 6b uses a much larger font than found anywhere else in the manuscript.
Answer: We thank the reviewer for constructive comments to improve the quality of our manuscript. The formatting errors are now corrected in table 1 and table 6 and we hope we have identified others throughout the paper.
- The paper measures SAM as one of the variables. Do you have the ability to also measure SAH as the SAM/SAH ratio has been found to be different in some other conditions as compared to controls? Given the link between SAM/SAH and DNA methylation this would be an important observation.
Answer: In the current study, we found that SAM has a tendency to deplete in the CSF of AD patients compared to controls. We agree with the reviewer about the importance of the ratio of SAM/SAH, as it is frequently used as an indicator of cellular methylation capacity, whereby a decrease in this ratio predicts reduced cellular methylation potential. Folate is integrally involved in both substrate synthesis and product removal via its role in methionine synthesis from homocysteine. We found a decrease in SAM probably reflecting the lack of methylation of homocysteine to methionine, a consequence of the abnormal SNP in MTHFD1. This may compromise SAM-dependent methylation reactions, while our data also suggests that homocysteine is bypassing SAH and going directly to glutathione via cysteine. Unfortunately, we are unable to measure levels of SAH to establish a change in the SAM/SAH ratio but hope that the data are sufficient in supporting the idea that homocysteine is being shunted to glutathione.
- The paper would benefit from having a bit more detail on the analysis used. While the methods are reasonably standard, it would not hurt to provide more information than is currently in the paper.
Answer: Unfortunately some of the methods are bespoke to the companies we used so the details are not available to us. DNA extraction and SNP analysis was carried out by LGC Genomics Ltd who do not provide details of their in–house methods. LifecodeGX also use a bespoke system to enter SNP data into their metabolic profiling for clients. This has been made clear in the methods and readers can consult directly with the companies for advice. The statistical analyses are well recognised methods used in genetics research so we felt it inappropriate to waste publication space on these details.
- The paper contains a number of typographical/grammatical mistakes that should be corrected, e.g., but not limited to, "Alzhiemer" in Figure 2b, "no significantly" instead of "not significantly" in the legend of Table 6, "This was not found to significantly difference" in the legend of Table 1, etc.
Answer: We thank the reviewer for constructive comments to improve the quality of our manuscript. The typological errors are now corrected throughout the manuscript and we hope we have captured all of them.

Reviewer 2 Report
The manuscript is focused on cerebral folate metabolism which represents be a key player in the onset of the Alzheimer’s disease. Different life events may decrease the drainage capacity e.g. infection, inflammation, trauma, etc and strategies to maintain/increase drainage may present an effective target for prevention and treatment of such a condition.
The work is interesting since it investigates the changes in metabolic profile which may provide early markers of at-risk individuals.
The language is readable although it may be somewhat improved.
Abstract: The aim of the study should be clearly stated. Full description for CSF is not provided.
Section 2.4. In general, no sufficient statistical analysis has been done for confirming the aims. Many uni and multivariate statistical analysis are available which could be used.
- Table 1, Cases used in the study: the two Tables are superimposed.
Author Response
- The language is readable although it may be somewhat improved.
Answer: We thank the reviewer for their constructive comments to improve the readability and understanding of our manuscript. The language has now been improved and we hope this has improved the flow.
- The aim of the study should be clearly stated in the abstract.
Answer: The aim of the study is now included in the abstract please see page 1, lines 17-18 as well as at the end of the introduction, lines 133-137
- Full description for CSF is not provided in abstract.
Answer: The full description of CSF is now included please see page 1, line 14. And is further described in the introduction lines 86-94
- Section 2.4. In general, no sufficient statistical analysis has been done for confirming the aims. Many uni and multivariate statistical analysis are available which could be used.
Answer: We believe that the statistical analyses do support the aims in demonstrating significant association of gene SNPs with Severe Alzheimer’s disease or normal aging. We have included further data analysis in additional tables for supplementary data, supplementary table s2, s3 and s4.
- Table 1, Cases used in the study: the two Tables are superimposed.
Answer: The formatting is now corrected.
Round 2
Reviewer 2 Report
The authors have adequately addressed all suggested remarks.